# Do dogs rationally infer the causes of failed actions?

**Amalia P. M. Bastos**[1,2*], **Gavin R. Foster**[1,3,4], **Patrick M. Wood**[1,2],
**Christopher Krupenye**[1]

**1** Department of Psychological & Brain Sciences, Johns Hopkins University, Baltimore, Maryland, United States of America, **2** School of Psychology & Neuroscience, University of St Andrews, St Andrews, United Kingdom, **3** Department of Philosophy, Carleton University, Ottawa, Ontario, Canada, **4** Department of Philosophy, Purdue University, West Lafayette, Indiana, United States of America

* apmb1@st-andrews.ac.uk

## Abstract

Humans regularly reason about the causes of events and actions we observe in the world, both to infer the physical properties and mechanisms of objects, and to understand others' actions. Evidence for causal reasoning in nonhuman animals is mixed, and may be more easily detected in some contexts than others. Dogs, for example, fail at most tests of causal reasoning pertaining to physical cognition, yet possess sophisticated sociocognitive abilities. In this pre-registered study, we test whether dogs are capable of making rational inferences about the causes of failed actions in two analogous experiments, which differed only in the nature of said failures. Dogs observed human agents either succeed or fail to open two gates, in contexts where their failures could be attributed either to the lack of competency of an agent, or the physical properties of a gate. If dogs are capable of making causal inferences equally in social and physical contexts, they should succeed in both experiments. However, if dogs are more likely to make social rather than physical causal inferences, they should find the competency context more interpretable than the physical one. Dogs failed to make rational inferences in either context, raising theoretical and methodological questions for future work.

## Introduction

The ability to rationally infer the causes of failed actions is an essential feature of human causal cognition. Suppose one watches an individual repeatedly fail to push open a door: there are multiple possible causes for this failure. On the one hand, the failure might reflect a lack of competency of the individual—perhaps they are too weak, or fail to recognize they are pushing on a door that requires pulling. Conversely, the failure might reflect an issue with the door itself—perhaps the door is jammed, or has been locked from the inside. There is strong evidence that the

**Data availability statement:** All data and code are provided as a supplementary material.

**Funding:** This project was made possible through the support of a grant from Templeton World Charity Foundation (CK) and a CIFAR Azrieli Global Scholar award (CK). APMB was supported by the Johns Hopkins Provost's Postdoctoral Fellowship Program. GF was supported by a SSHRC Graduate Scholarship and a Michael Smith Foreign Study Supplement.

**Competing interests:** The authors declare no competing interests.

ability to infer which cause underlies a failed action emerges early in humans [1]. In humans, this ability plays a role in enabling us to predict future outcomes [2,3], choose the most effective partners for cooperation [4,5], and determine the best course of action to remedy a failure [5–7]. The ability to reflect on one's own failed action and rationally infer its cause is fundamental to satisfying our goals in the world. However, it remains an open question as to whether this capacity is uniquely human or also present in nonhuman animal species.

Dogs (*Canis familiaris*) are unique among animals in having undergone an exceedingly long domestication process with humans, beginning around 30,000–40,000 years ago [8]. Some research suggests that this domestication process has played a significant role in influencing and shaping their cognition [9–11], and perhaps has led to dogs' unique sensitivity to human social cues. For example, dogs are adept at detecting and identifying basic human emotions [12–14], and can utilise a variety of human social cues like pointing and gaze direction to both predict human behaviour and infer communicative intent by humans, particularly in two-choice tasks [15–18]. Dogs use human gaze direction to succeed in some perspective taking paradigms [19–22], and can rapidly learn novel social cues after only minimal exposure [23].

However, despite their particular sensitivity to human social cues, dogs appear less adept at causal ('means-end') reasoning than some other species, such as nonhuman primates [24–26]. Some studies suggest that dogs are not sensitive to the function of various objects (e.g., the tugging of a rope) even after training [27,28]. Other research suggests that dogs can exhibit causal reasoning on similar tasks only after extensive training and only in particular contexts [29,30], although it is more difficult to determine the extent to which performance following extensive training reflects cognitive mechanisms supporting causal inference, or a generalisation of simpler behavioural heuristics that are equally applicable across operationally similar tasks. One possibility is that dogs' poor performance in many causal inference tasks reflects a limited understanding of the physical world (such as the properties of objects), but that they might be able to reason causally in more social contexts. Studies suggest that dogs are sensitive to human competency across several familiar contexts, tracking the identity of individuals who are successful at performing certain actions and those who are not [31,32], but there is little evidence to suggest that they have a causal understanding of competency. Alternatively, it remains possible that dogs exhibit a more general lack of causal understanding across both social and physical contexts. That is, although their evolutionary history of domestication and early learning environments lead them to adeptly rely on human social cues to make predictions and guide their own behaviour, even in social contexts they lack an ability to interpret events in terms of their underlying causal structure. If this were the case, we should expect to find a more general inability to engage in causal reasoning in both social and physical contexts across both dogs and wolves, whose performances in physical cognition tasks are comparable and therefore seem to be largely unaffected by the domestication process in dogs [28,33,34 *but see* 35].

To determine whether dogs can rationally infer physical and social causes for failed actions, here we conceptually replicated a study by Gweon & Schulz [1], which showed that infants discriminate between failures in the world that are caused by object-based features and those caused by agential features related to the competency of a human actor. In one of their studies, infants were presented with one of three scenarios in which experimenters attempted to operate a green toy (pushing a button to produce a sound) across four trials: either (a) a single experimenter succeeded twice and failed twice, (b) two experimenters each succeeded once of failed once, or (c) one experimenter succeeded in two trials while the other failed in two trials. Infants then failed to elicit the sound on the green toy. In the first two scenarios, where both experimenters had the same success rate, infants who failed to elicit the sound were more likely to reach for a new toy, as if attributing their failure to the toy which they had observed often failing to function across users. However, in the third scenario where infants had observed some agents reliably eliciting a sound and others reliably failing to do so, infants were more likely to request help by handing the toy to their parent. The authors interpret these results to indicate that infants could rationally infer whether the toy's failure to function owed to an issue with the toy itself or to the incompetence of the agents attempting to operate it.

Analogously, here we present dogs with two experiments involving a highly familiar context: humans opening pet gates. In each experiment, two human experimenters took turns attempting to open two different pet gates. Then, at test, the experimenters stood beside different gates, while the dogs' owner called the dog from behind the barrier; dogs could choose which experimenter/gate to approach to reach their owner. In Experiment 1, one (competent) experimenter successfully opened both gates, while another (incompetent) experimenter failed to open both. In Experiment 2, both experimenters successfully opened one gate but failed to open the other, suggesting that the issue stemmed from the gate itself. If dogs rationally infer both social and physical causes for failure, in Experiment 1 they should approach whichever gate the competent experimenter was standing next to, whereas in Experiment 2 they should approach whichever gate was successfully opened by both experimenters. If dogs' causal inference abilities are confined to the social realm, they should succeed in Experiment 1 but not Experiment 2. A more general lack of causal reasoning would predict failure in both experiments.

Crucially, although both human competence and gate functionality are needed for the gate to be opened, each of the two experiments manipulated only one of these factors. In doing so, we directly investigated whether dogs are sensitive to differences in human competency, object functionality, both factors, or neither factor.

## Methods

### Subjects and apparatus

Our subjects for Experiment 1 were 22 dogs, one of which was excluded due to experimenter error, and another for jumping the gates, resulting in a final sample of 20 dogs (12 females; age: M = 4.1 years, SD = 2.2; see Table 1). For Experiment 2, we recruited 20 dogs (15 females; age: M = 4.0 years, SD = 2.4; see Table 2). Our sample sizes for both studies were determined by a pre-registered stopping rule (link: https://aspredicted.org/mbyv-3kkc.pdf), whereby we began with an initial sample of 20 dogs and included additional subjects as needed until we reached a Bayes Factor greater than 3 or smaller than 0.33m [36] for our one-sample t-test analyses with an even number of subjects. All dogs were family pets in the Baltimore area (USA) recruited for participation in the study through an online survey. Dogs were tested in the canine cognition testing suite of the Canine Minds Collaborative, on the Homewood campus of Johns Hopkins University. This work was carried out under the approval of the Johns Hopkins University Animal Care and Use Committee (reference number DO23A102).

Dogs were only recruited for the study if they could be recalled by their owners and did not exhibit any fearful responses to indoor fencing or gating. This was confirmed when they arrived at the laboratory, when dogs were allowed to freely explore the testing room prior to the experiment, and in this case both gates were left open and dogs were encouraged to move through the two gates by their owners, twice clockwise and twice anticlockwise.

Table 1. Demographic information for all subjects in Experiment 1. Ages are given in years. Breed size categories were determined as per American Kennel Club (AKC) designations, either for the dog's primary breed (if purebred) or for the component breed the dog most resembled in terms of size (for mixed breed dogs). Size categories as per the AKC are giant (75+pounds), large (55-85 pounds), medium (35-65 pounds), small (7-35 pounds), and toy (2-9 pounds). Mixed breed information was provided by the owners. Dogs for whom breed mix breakdowns were obtained through commercially available genetic testing kits are indicated by an asterisk.

| Subject | Name | Sex | Age | Breed Size Category | Breed |
|---|---|---|---|---|---|
| 1 | Sandy | F | 3 | Large | Golden Retriever |
| 2 | Watson | M | 3 | Large | Golden Retriever |
| 3 | Hudson | M | 2 | Large | Golden Retriever |
| 4 | Indy* | F | 6 | Medium | *Mixed Breed:* Australian Cattle Dog x Treeing Walker Coonhound x Boxer x Norwegian Elkhound |
| 5 | Koji | M | 4 | Small | Miniature American Shepherd |
| 6 | Hope | F | 6 | Large | Labrador Retriever x Boxer |
| 7 | Cully | F | 3 | Medium | Border Collie |
| 8 | Dutch | M | 1 | Small | Whippet |
| 9 | Scout | F | 9 | Medium | Airedale Terrier |
| 10 | Jax | M | 3 | Large | German Shepherd Dog |
| 11 | Luna | F | 4 | Medium | *Mixed Breed:* Labrador Retriever x American Staffordshire Bull Terrier |
| 12 | Laney | F | 3 | Medium | *Mixed Breed:* Bull Terrier x American Pit Bull Terrier |
| 13 | Razor | F | 5 | Large | Labrador Retriever |
| 14 | Kamden | M | 2 | Medium | *Mixed Breed:* Labrador Retriever x American Stafforshire Bull Terrier |
| 15 | Logan | M | 4 | Large | Labrador Retriever |
| 16 | Sophie | F | 3 | Medium | *Mixed Breed:* Labrador Retriever x Border Collie |
| 17 | Hadley* | F | 7 | Medium | *Mixed Breed:* American Foxhound x American Bulldog x Bulldog |
| 18 | Gidget | F | 2 | Large | *Mixed Breed:* Old English Sheepdog x Standard Poodle |
| 19 | Sawyer | M | 9 | Medium | *Mixed Breed:* English Staffordshire Terrier x Australian Cattle Dog |
| 20 | Ripley | F | 3 | Large | Standard Poodle |

Both studies took place in a rectangular room (4.9 m x 3.2 m), where one end of the room (1.2 m x 3.2 m) was fenced off. The fencing included two lockable pet gates (EveryYay steel gates) which could be swung open to form an opening measuring 46 cm x 69 cm, which was sufficiently large for all participants to pass through, as dogs have been shown to avoid openings that are too small for their body size [37]. The two gates were located equidistant from either side of the room and from an equilateral triangular fence divider between them (0.6 m on each side). A line of tape on the floor crossed the width of the room from the tip of the triangular fence, demarcating a distance of 0.6 m in front of each of the gates. This distance was chosen as it was slightly beyond the full extension of the gate when it was swung fully open, and so we considered any approach within the range of the open gate to be a choice for that side. A chair was positioned behind the triangular fencing, equidistant to both gates, against the back wall of the room. The floor was marked with tape to indicate the standing positions of the two experimenters performing the demonstrations (hereafter, E1 and E2) and the assistant handling the dog (hereafter, the handler).

### Training and procedures

In each of our two experiments, dogs were first allowed to investigate the study room in their own time, then encouraged to move through each gate twice in both directions. Following this habituation, the owners were briefed on the general procedure of the forthcoming test trials, but given no information on the exact actions that would be performed by any of the experimenters. Owners were instructed to look at a mark on the wall on the opposite side of the room throughout the trial, except for the moment when they recalled their dog, at which time they were told to look straight ahead at their dog and then return their gaze to the wall (such that they were not recalling their dog to either side of the triangle in front of

**Table 2. Demographic information for all subjects in Experiment 2.** Ages are given in years. Breed size categories were determined as per American Kennel Club designations, either for the dog's primary breed (if purebred) or for the component breed the dog most resembled in terms of size (for mixed breed dogs). Size categories as per the AKC are giant (75+ pounds), large (55-85 pounds), medium (35-65 pounds), small (7-35 pounds), and toy (2-9 pounds). Mixed breed information was provided by the owners. Dogs for whom breed mix breakdowns were obtained through commercially available genetic testing kits are indicated by an asterisk.

| Subject | Name | Sex | Age | Breed Size Category | Breed |
|---|---|---|---|---|---|
| 1 | Carly | F | 10 | Large | *Mixed Breed:* Labrador Retriever x Border Collie |
| 2 | Piper | F | 3 | Medium | Australian Shepherd |
| 3 | Rosie | F | 3 | Large | Miniature Poodle x Golden Retriever |
| 4 | Odin | M | 1 | Large | Boxer |
| 5 | Tucker | M | 4 | Giant | Great Pyranees |
| 6 | Miller | M | 3 | Large | Golden Retriever |
| 7 | Renny | F | 3 | Small | Basset Fauve de Bretagne |
| 8 | Kota | F | 3 | Medium | Border Collie |
| 9 | Bonnie* | F | 9 | Medium | *Mixed Breed:* Australian Cattle Dog x Australian Shepherd |
| 10 | Nyx* | F | 6 | Medium | *Mixed Breed:* Siberian Husky x American Pit Bull Terrier |
| 11 | Tela | F | 2 | Medium | *Mixed Breed:* Labrador Retriever x Whippet |
| 12 | Kudos | M | 3 | Medium | Australian Shepherd |
| 13 | Beatrice* | F | 3 | Medium | *Mixed Breed:* Border Collie x Chow Chow |
| 14 | Storm | F | 5 | Large | German Shepherd Dog |
| 15 | Buster | M | 6 | Small | Beagle |
| 16 | Ella | F | 3 | Medium | Australian Shepherd |
| 17 | Rosita | F | 2 | Large | Dobermann |
| 18 | Marble | F | 2 | Medium | American Foxhound x American Staffordshire Terrier |
| 19 | Mabel | F | 2 | Large | Golden Retriever |
| 20 | Tess | F | 7 | Small | Parson Jack Russell Terrier x Dachshund |

them, but straight towards them). Throughout the entire trial, they were also instructed to sit with both hands on their lap and not move their hands or body.

Each dog participated in two trials. Trials started with the owner sitting in the chair behind the fence, with both gates locked shut, and E1 and E2 standing on different sides of the rooms. The handler entered the room with the dog on leash and the door was closed behind them by another researcher (hereafter, the assistant). The trial began once the handler and the dog were in their starting positions and the door was closed behind them.

Each trial involved two demonstrations by each experimenter. A demonstration consisted of an experimenter moving towards one of the gates and calling the dog's name twice while clapping their legs, then either successfully opening it, followed by a "Yay!" vocalization and both hands by their legs with open palms towards the dog (and then closing the gate), or shaking the gate and failing to open it, followed by a "Aww" vocalization accompanied by the same hand gesture. Once the first experimenter had attempted to open both gates, they returned to her starting position and the second experimenter took their turn attempting to open both gates. The order in which they approached each gate, and which of the two experimenters went first, was pseudorandomised and counterbalanced between dogs (see S1 File).

In Experiment 1, to assess whether dogs can infer that gate-opening failures were due to the competencies of the two experimenters, one experimenter consistently failed to open each gate in turn, while the other consistently succeeded to open both gates (a schematic representation is shown in Fig 1).

In Experiment 2, to assess whether dogs can infer that failures to open one gate but not the other were likely due to a physical difference between the two gates (such as one of them being broken), both experimenters failed to open the same gate and succeeded at opening the other (Fig 2).

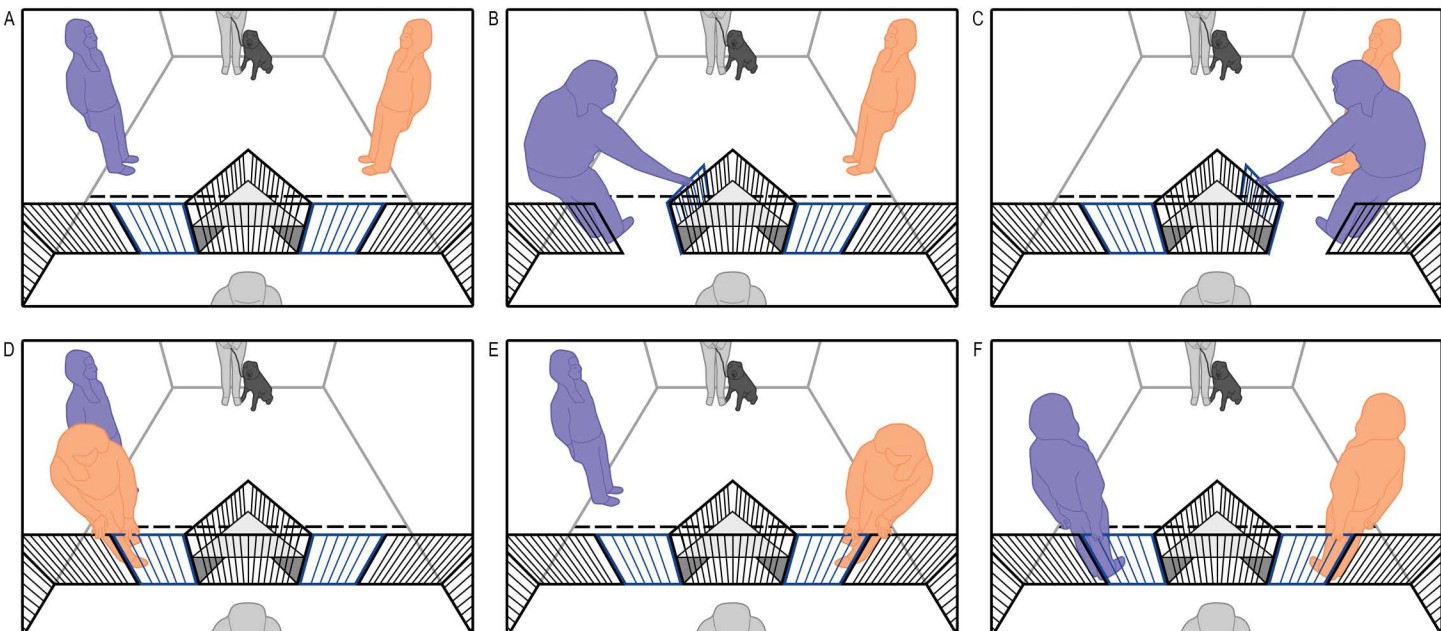

**Fig 1. Schematic diagram for the events in each trial of Experiment 1.** The handler is depicted with the dog at the top of each frame, and the owner is depicted seated behind the fence at the bottom of each frame. Each trial started with the dog positioned in the middle of the room and both experimenters positioned to either side **(A)**. The competent agent opened both gates in turn **(B, C)**, and the incompetent agent attempted to but failed to open the same two gates **(D, E)**. The order in which agents performed their demonstrations, and which gate each one moved to first, were counterbalanced. Once both agents performed their demonstrations, they approach their gates simultaneously and each stood with one hand on their gate, at which point the owner recalled their dog and the dog was let off the leash to choose which gate to approach. The side of dogs' first approaches were coded as the side where dogs first stepped beyond a taped line 0.6m in front of the gates, and the amount of time spent near each of the two agents was coded as the amount of time dogs spent within that space on the right or left side.

Following four demonstrations (two by each experimenter, such that each experimenter approached and interacted with each of the two gates once), E1 and E2 moved next to separate gates, with each experimenter placing one hand on their gate. The handler then prompted the owner to recall their dog to them, and unclipped the dog's leash. The dog was given 15 seconds to move anywhere in the room. We expected that, if dogs can rationally infer both causes of failed actions in our studies, then they should preferentially move toward the gate closest to the competent experimenter in Experiment 1 and move toward the gate that was successfully opened by both experimenters in Experiment 2.

At the end of the 15 second choice period, the assistant opened the door to the room and called the dog out before closing the dog behind it. Once the dog left the room, the experimenters let the owner out from behind the fence and then came out of the room to greet the dog outside. This ensured that dogs did not observe their owner move through either gate, which might influence their choices in the following trial. While the dog was outside of the testing room, the experimenters set up for the second test trial, which was identical to the first except that the two experimenters switched sides.

## Analyses

All trials were recorded on video from three angles. Videos were annotated by a trained coder who was blind to experimental hypotheses. This individual coded the first side chosen by the dog (as defined by a paw partially or fully crossing the line of tape running across the front of both gates), as well as the amount of time spent in front of each option, again as demarcated by the line of tape. All coding was performed in BORIS v8.27 [38]. Across experiments, when making a choice, dogs were approaching both a gate and a human experimenter but since we manipulated (and were interested

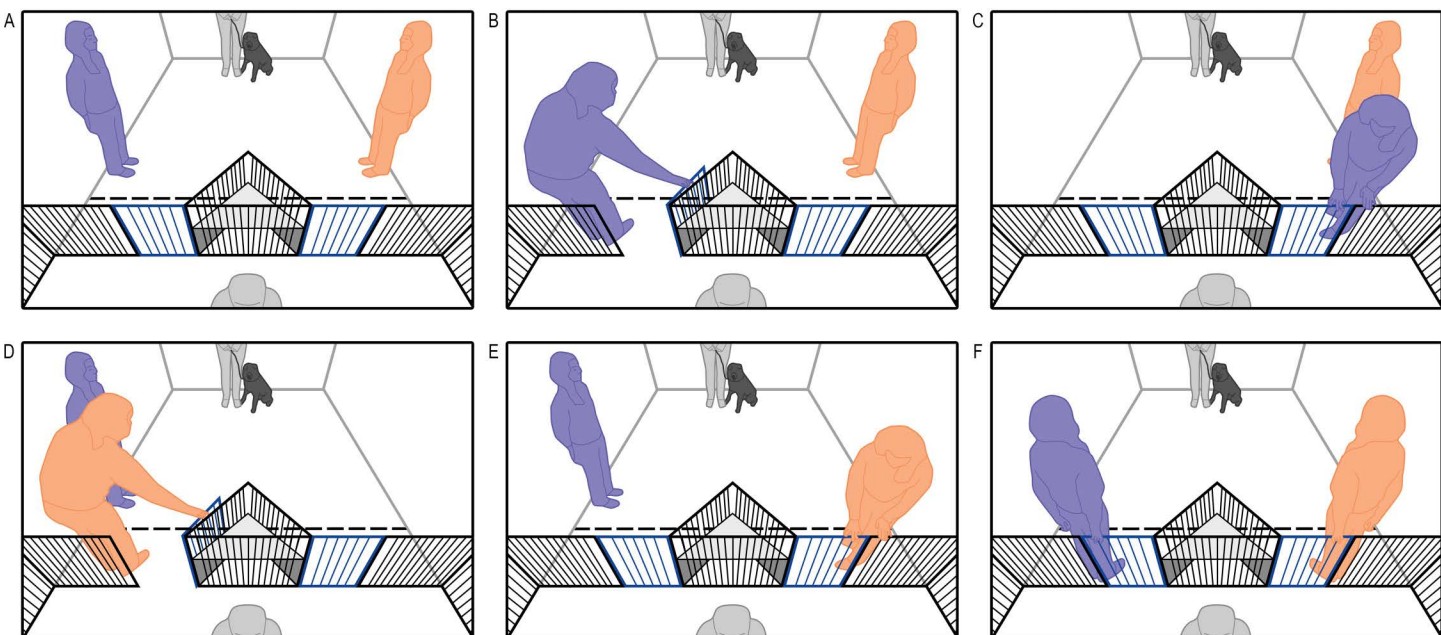

**Fig 2. Schematic diagram for the events in each trial of Experiment 2.** Again, trials started with the dog positioned in the middle of the room and the two experimenters positioned to either side **(A)**. The first experimenter took their turn successfully opening one of the two gates but trying and failing to open the other **(B, C)**. The second experimenter performed the same actions towards the same two gates **(D, E)**. The order in which experimenters performed their demonstrations, and which gate they approached first, was counterbalanced. After demonstrations, experimenters moved simultaneously to their gates and placed a hand on their gate, then the owner recalled their dog as it was let off the leash to approach. The side of dogs' first approaches were coded as the side where dogs first stepped beyond a taped line 0.6m in front of the gates, and the amount of time spent near each of the two gates was coded as the amount of time dogs spent within that space on the right or left side.

in dogs' sensitivity to) social competence in Experiment 1 and physical functionality in Experiment 2, we describe dogs' choices as approaches to agents in Experiment 1 and to gates in Experiment 2.

As outlined in the pre-registration for this study (link: https://aspredicted.org/mbyv-3kkc.pdf), our primary analysis was a Bayesian one-sample t-test comparing the proportion of trials for which dogs first approached the predicted agent or gate (averaged across both trials) at the group level, compared to a chance level of 0.5 success (1 of 2 trials correct), for each of the two experiments. To further characterise our results, we used a Bayesian mixed effects model to investigate dogs' choices in each of the two experiments, described by the equation: Correct ~ (1 | Subject), where "Correct" is a 0 or 1 value indicating the dog's choice of the causally predicted agent (Experiment 1) or gate (Experiment 2) in any given trial. We obtained Bayes factor values for this model by comparing it to the null model Correct ~ 1. Adding trial number as a random effect did not affect the results of either model.

We additionally investigated the amount of time dogs spent at each of the two options (agents or gates) in each experiment using a Bayesian mixed-effects model. The dependent variable was the proportion of time spent at the predicted agent (Experiment 1) or gate (Experiment 2) over the total amount of time spent near both options, and the models are described by the equation: Proportion of Time ~ (1|Dog). Models of the proportion of time dogs spent at the two agents or gates were also compared against null models described as Proportion of Time ~ 0.5, to obtain a Bayes factor value in support of the alternative hypotheses. As before, trial time did not affect either model.

Post-hoc exploratory models also investigated whether dogs showed a preference to approach the gate an agent most recently opened (Experiment 1), as described by the equation Side Approached ~ Gate Last Opened + (1|Dog). We also

investigated whether dogs showed a preference to approach the gate most recently touched (both Experiment 1 and Experiment 2), as described by the equation: Side Approached ~ Gate Last Touched + (1|Dog).

All analyses were carried out in R [39] using the brms [40] and BayesFactor [41] packages.

## Results

### Experiment 1

Dogs were no more likely to approach the competent agent than the incompetent agent in the first experiment (multi-level logistic regression; 0.22, 95%CI[−0.49, 0.98]), and there was considerable variability between dogs in their choices (0.62, 95%CI[0.02, 1.82]). Dogs' approaches to the competent agent were no more likely than approaches to the incompetent agent when these approaches were characterised as proportional correct choices (Bayesian one-sample t-test: BF = 0.289, N = 20). A comparison between the logistic regression model for binary approaches and a null model revealed moderate evidence for the null hypothesis (BF = 0.282). This was true even if trial order was accounted for (BF = 0.157).

We also investigated whether dogs spent more time near the competent agent than the incompetent agent, as defined by having any body part beyond the line in front of the gate on that side (i.e., within 0.6m of said agent). The proportion of time dogs spent near the competent agent, relative to the total time they spent near either agent, varied considerably across dogs (logistic regression standard deviation: 0.18, 95%CI[0.01, 0.54]; Fig 3), but dogs were no more likely to spend time near the competent agent (multilevel logistic regression; 0.07, 95%CI [−0.30, 0.45]). A comparison between the alternative hypothesis and null models revealed strong evidence in favour of the null hypothesis (BF = 0.082).

Post-hoc analysis also did not reveal other overarching patterns to dogs' choices in this experiment. Dogs did not exhibit a preference for the gate last opened (multilevel logistic regression: 1.12, 95%CI[−0.08, 2.74]), and this was not

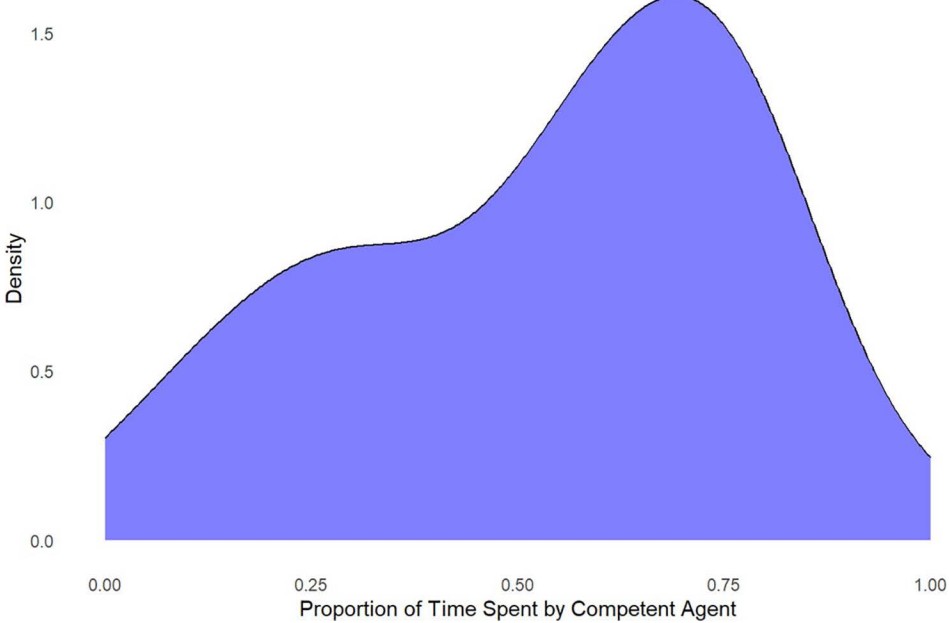

**Fig 3. Density plot for the time dogs spent in the vicinity (defined as within 0.6m) of the competent agent, as a proportion of the total amount of time spent in the vicinity of either of the two agents.**

affected by the side of the last opened gate (−0.77, 95%CI[−2.85,1.07]). Dogs also did not exhibit a preference for the gate last touched (0.78, 95%CI[−0.34, 2.17]) and this was also not affected by gate side (0.12, 95%CI[−1.53, 1.81]). Twelve of twenty dogs first approached the same side in both trials, and there were 26 approaches to the agent on the right-side gate from a total of 40 approaches across all dogs and all trials, suggesting there was no population-wide bias for either side of the room.

## Experiment 2

We found no evidence that dogs were more likely to approach the functional gate in both trials relative to chance (Bayesian one-sample t-test: BF = 0.196, N = 20). This was also true when trials were considered individually (multilevel logistic regression: −0.54, 95%CI[−1.70,0.43]). The model also revealed considerable variation in choices between dogs (standard deviation: 1.41, 95%CI[0.09, 3.61]). A comparison between the model investigating approach to the functional gate and a null model provided anecdotal support for the null hypothesis both when trial was considered as a random effect (BF = 0.373) and when it was not (BF = 0.955).

As with Experiment 1, we investigated the proportion of time dogs spent in proximity to either gate (defined as within 0.6m of each gate; Fig 4). We found no evidence to suggest that dogs spent more time at the functional gate compared to the non-functional gate (multilevel logistic regression: −0.18, 95%CI[−0.74,0.36]), with a comparison to the null model providing anecdotal support in favour of the null hypothesis when trial was not included as a random effect (BF = 0.701), and moderate support when it was included (BF = 0.202).

Exploratory analyses revealed that dogs were no more likely to approach the gate last touched by an agent before the time when they were allowed to make a choice (multilevel logistic regression: 0.13, 95%CI[−1.25,1.57]), although including the gate last touched as a fixed effect provided a moderately better fit for the data than the null model (BF = 4.083).

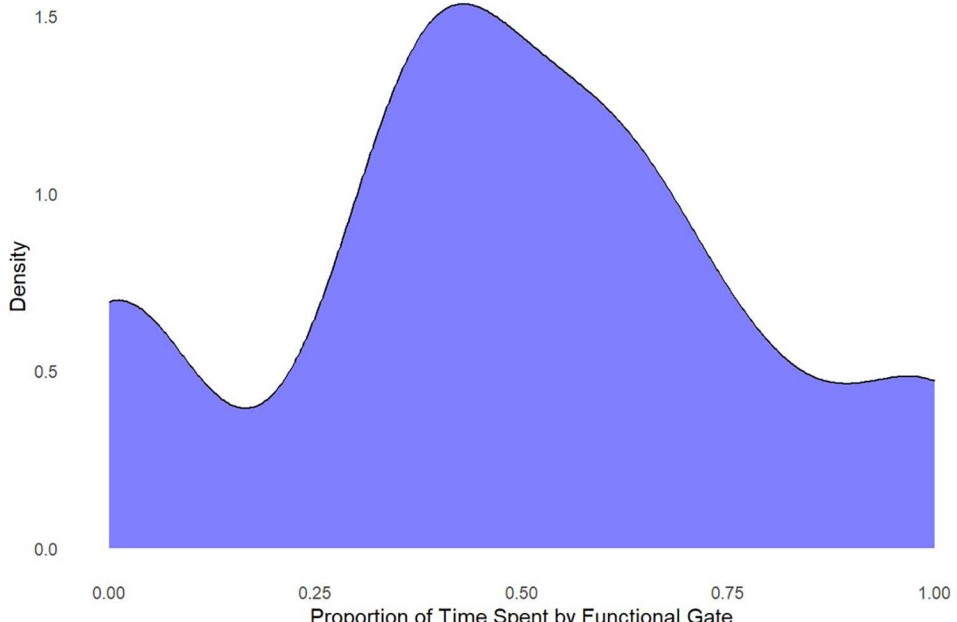

**Fig 4. Density plot for the time dogs spent in the vicinity (defined as within 0.6m) of the functional gate, as a proportion of the total amount of time spent in the vicinity of either of the two gates.**

In terms of side preferences, ten of twenty dogs selected the same side on both trials, and the right-side gate was approached in 18 of 40 trials across all dogs, suggesting no population-wide preferences to approach either side of the room.

## Discussion

In this study, dogs did not behave as if they were rationally inferring the causes of failed actions of either social or physical nature. Dogs did not preferentially approach, or spend time near, an experimenter who was competent at opening both gates, compared to one that tried to but failed to open the same gates. Dogs also did not make more first approaches to, or spend longer next to, a functional gate that both experimenters could successfully open, relative to a non-functional one that could not be opened by either experimenter. This suggests that, at least in this context, dogs did not make causal inferences about the reason for the gates being opened successfully or not.

Dogs' first approaches and proportional time spent by either of the two agents (Experiment 1) or gates (Experiment 2) were random. The last gate opened, last gate touched, and side of gate could not explain dog's choices, suggesting that dogs were not using some other heuristic strategy or bias to make their choices. It is unlikely that this failure was due to a general lack of attention. We intentionally positioned the dog's owner behind the gates and across the room from their dog, so that dogs would look toward the gates throughout trials. During demonstrations, experimenters always called the dog's name before interacting with each gate. All dogs attended to every demonstration, as experimenters only carried out their actions once the dog looked toward the experimenter calling them. Demonstrations were accompanied by salient hand gestures and vocalisations relating to angry/frustrated and happy emotional content, which dogs can discriminate between [12,13,18,42]. It is particularly interesting that, despite the saliency of the demonstrations, and the fact that the emotional valence of said demonstrations should be discriminable to dogs, dogs still failed this task on all accounts.

Dogs' failure at this task also cannot be attributed to a lack of motivation. All dogs that participated in the study were familiar with a recall cue (e.g., "come", "here") and attempted to reach their owner in every trial. Therefore, all dogs approached one of the two gates after being recalled by their owners, and most dogs alternated between the two gates once they realised they could not reach their owner immediately following their first approach. Further, one might argue that task complexity in our study was relatively low, posing fewer demands than the original infant study it was based upon [1]. Whilst infants in the original study by Gweon and Schulz [1] acted upon two objects in order to determine the causality of different actions, dogs only passively observed the causal nature of experimenters' interactions with two gates and did not need to test out causal hypotheses by interacting with said gates themselves. On the other hand, it is possible that increasing dogs' agency to intervene on causal situations or test causal hypotheses might enhance their performance, perhaps by improving their understanding of the mechanisms at play or by increasing the constellation of possible behaviours dogs might exhibit in response to objects associated with causally unexpected outcomes. For example, in a recent study by Völter and colleagues [43], dogs spent more time interacting with an object that seemingly disappeared and thus violated spatial continuity expectations, than an object that did not.

Further, it is unlikely that dogs failed at this task because the actions they observed were not ecologically relevant. The gate opening context was specifically chosen because it was likely to be very familiar and salient to dogs. Dogs observe humans open and close doors daily, often following humans through said doors. This often includes observing the opening and closing of, and moving through, pet gates exactly like those used in the experiment, which are common in pet homes, dog daycares, and training schools. Therefore, one might expect that, if dogs are capable of causal reasoning, they should be able to apply it to such a familiar context. Nevertheless, it is still possible that this context may have been unfamiliar to some of the subjects in the study.

It is also possible that dogs are not attentive or sensitive to the functionality of gates, given that, although they can learn to do so [44], dogs don't typically open gates themselves. If that is the case, then they may not have realised that both agents failed or succeeded to open one particular gate in Experiment 2. If that was the case, then either dogs may have

perceived this task as choosing between two equally competent humans or they may not have noted any appreciable difference between either option. Either of these alternative hypotheses would predict that dogs should approach either of the gates at chance. Importantly, however, the former hypothesis would also have predicted that dogs would have discriminated between the competency of the two agents in Experiment 1, which they did not.

Nevertheless, it might have been beneficial to provide additional scaffolding through which subjects could have understood how the gates worked. For example, we could have included additional demonstration at the start of each trial where an additional experimenter explicitly demonstrates the functionality of the gates to the dogs, perhaps by opening one gate to allow the dog access to their owner on the other side, but not the other. This familiarisation might allow any dogs that are unfamiliar with gates to learn that they could be opened for them. Another option would be to allow subjects to pass through the gates once the experimenters opened them during the experimental demonstrations. Although we considered this possibility in the design of this study, we refuted it on three grounds: first, this would increase the length of time between demonstrations and moment when dogs made their choice at the end of the trial, potentially increasing existing cognitive load for the task; second, given the proximity of the gates to the owners, this may already provide dogs with proximity to their owners halfway through the trial and therefore reduce their motivation to approach them at the end; and third, and most importantly, differentially experiencing moving through either gate might lead dogs to make the correct choice at test not because of the competency of the agents, but because they learned to associate the "competent" agent with proximity to their owner in the social task, or the side of the "functional" gate with proximity to their owner in the physical task. In the physical task in particular, it is likely that this association would have led to a perseveration error, which dogs acquire after very few repetitions [45–47].

Alternatively, one could also envision an alternative to Experiment 2 where both gates are demonstrated by a single experimenter, who is repeatedly successful at operating only one gate but not the other. This would eliminate the dogs' need to attend to the competencies of two different humans and from that infer the cause of their failure, as infants did in the Gweon and Schulz study [1], but it might have made the physical properties of the two gates more salient to subjects. Another option would be to train dogs to push gates open and then let them experience the functionality of both gates firsthand, given that previous research has shown that dogs can differentiate between "heavy" and "light" swinging doors, but only after they experience the two door types themselves [44].

Given that pet dogs are familiar with human social partners performing actions for them (including opening doors) and with pet gates, we anticipated that this task would be highly ecologically relevant for our subjects. However, since both our experiments require dogs to integrate both social and physical cognition to determine the cause of human or gate opening failures, it could be that these demands proved too challenging. Therefore, it is possible that although dogs may be capable of making inferences about causes for social or physical failures individually, the cognitive demands required to integrate information across both these domains in the context of our task may have been too high.

It is possible that the presence of dogs' owners, and their calling of the dogs' names, could have been so salient to subjects that it overshadowed more subtle representations and choices dogs would have otherwise made in response to the demonstrations provided. It is difficult to determine from our data alone whether our demonstrations failed to elicit causal understanding in dogs because they are not cognitively equipped to make the inferences these manipulations required, or because some aspect of these manipulations made it too difficult for them to do so.

These findings stand in contrast to some of the existing literature suggesting that dogs can ascribe competency to humans [31,32], differentiate between contexts where human incompetence is either intentional or unintentional (as in the willing vs unwilling paradigm [48], and distinguish selfish and unselfish actors [49]. For example, in Chijiiwa et al. study [31], female dogs preferentially approached a human who successfully opened a jar to obtain a treat inside it. One possible reason for the difference between our findings and those of the Chijiiwa study might relate to cognitive load and movement. In the Chijiiwa study, experimenters remained stationary throughout the entire trial, potentially making it easier for dogs to track the last location where the jar was positioned, and therefore the competent person associated with it.

Alternatively, dogs' choices may have been mediated not by discriminating between human competencies, but by stimulus enhancement, and indeed several studies that control for stimulus enhancement while testing dogs' valuations of two different people have failed to find any effect [50–53].

Although there is evidence that dogs can discriminate between experimenters when they move around the room for similar lengths of time in other contexts [e.g., 52–55], the pseudorandomisation and counterbalancing of the manipulations in our study may have nevertheless made it too difficult for dogs to track the humans' actions, even if they are capable of attributing competencies to different human agents. The cognitive load and memory requirements to track the identities and locations of the two experimenters as they move around the room to perform their actions and then return to their original positions, whilst simultaneously demonstrating competing levels of competency at opening gates, may have been too great for the dogs. If the dogs' failure merely reflects a cognitive load or memory constraint, then it is still possible that dogs are in fact capable of representing and ascribing competency to different agents, and/or functionality to physical objects, but the methodology presented in this study made it impossible for subjects to perform as such. Future work should further address dogs' causal reasoning using different paradigms, to help determine if this failure is representative of dogs' cognitive capacities or if their failure at this task reflects unrelated issues of methodology or contextual variables.

If further work continues to demonstrate failure of causal reasoning in dogs, even when tasks are simplified to address the potential issues of stimulus saliency, cognitive load, and working memory capacity that we have raised, then that would suggest that dogs' success on sociocognitive tasks, including recent theory of mind tasks [e.g., 22], may not reflect a full-blown causal understanding of other minds but rather an evolved or learned response to relevant social cues. Indeed, this leaner interpretation is consistent with dogs' apparent failures in experience projection theory of mind paradigms where dogs cannot rely on differential cues across conditions [55,56]. If this interpretation is correct, we must also account for the results of the jar-opening study by Chijiiwa and colleagues' study [31]. The results of Experiment 1 suggest that in studies such as these, the dogs may be tracking something other than the competency of human agents, potentially experiencing some form of local or stimulus enhancement. However, given that in Experiment 2 dogs failed to track that it was mechanical failure that caused a particular gate to not open, there is doubt that dogs are even capable of tracking physical mechanisms as the source of causal failure. Such a conclusion is consistent with the many studies suggesting that dogs are less adept at physical and causal cognition [24,26], even compared to wolves [33, but see 57]. Future work should continue to probe dogs' capacity for causal reasoning across social and physical contexts, and whether a flexible sensitivity to social cues can exist in the absence of genuine causal understanding.

## Supporting information

**S1 File. Gates_Supp_Video_Small.mp4 – Supplementary Video 1.** Video demonstrating study procedures for Experiments 1 and 2.
(MP4)

**S2 File. R Markdown File Dog Inference Study.pdf – Supplementary Script 1.** R markdown script for statistical analyses carried out for the study and reported in the results section of the manuscript.
(PDF)

**S3 File. Gates_Exp1_Clean.csv – Supplementary Data 1.** Raw data file for Experiment 1.
(CSV)

**S4 File. Gates_Exp2_Clean.csv – Supplementary Data 2.** Raw data file for Experiment 2.
(CSV)

## Author contributions

**Conceptualization:** Amalia P. M. Bastos, Gavin R. Foster, Christopher Krupenye.

**Data curation:** Amalia P. M. Bastos.

**Formal analysis:** Amalia P. M. Bastos.

**Investigation:** Amalia P. M. Bastos.

**Methodology:** Amalia P. M. Bastos, Gavin R. Foster, Patrick M. Wood, Christopher Krupenye.

**Project administration:** Amalia P. M. Bastos, Gavin R. Foster, Patrick M. Wood.

**Supervision:** Christopher Krupenye.

**Writing – original draft:** Amalia P. M. Bastos, Gavin R. Foster.

**Writing – review & editing:** Amalia P. M. Bastos, Gavin R. Foster, Christopher Krupenye.

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
