## [Decision Letter · Decision Letter 0]

26 Oct 2025

PONE-D-25-40425Do dogs rationally infer the causes of failed actions?PLOS ONE

Dear Dr. Bastos,

Thank you for submitting your manuscript to PLOS ONE. After careful consideration, we feel that it has merit but does not fully meet PLOS ONE’s publication criteria as it currently stands. Therefore, we invite you to submit a revised version of the manuscript that addresses the points raised during the review process.

Dear authors,

After careful consideration by the reviewers, I am recommending a Major Revision of the original manuscript submitted to PLOS One. The reviewers appreciated the contributions of the study, even though the results were negative. However, they raised questions regarding the experimental design and requested an update of the bibliographic references. The detailed comments from both reviewers are specified below.

We look forward to receiving your revised manuscript.

Kind regards,

Maria Eduarda Lima Vieira

Guest Editor

PLOS ONE

Journal Requirements:

4. Please amend your authorship list in your manuscript file to include author Patrick Mitchell Wood.

5. Please amend the manuscript submission data (via Edit Submission) to include author Patrick M. Wood.

6. Please remove all personal information, ensure that the data shared are in accordance with participant consent, and re-upload a fully anonymized data set.

Additional guidance on preparing raw data for publication can be found in our Data Policy (https://journals.plos.org/plosone/s/data-availability#loc-human-research-participant-data-and-other-sensitive-data ) and in the following article: http://www.bmj.com/content/340/bmj.c181.long .

Additional Editor Comments :

Dear authors,

After careful consideration by the reviewers, I am recommending a Major Revision of the original manuscript submitted to PLOS One. The reviewers appreciated the contributions of the study, even though the results were negative. However, they raised questions regarding the experimental design and requested an update of the bibliographic references. The detailed comments from both reviewers are specified below.

Reviewers' comments:

Reviewer's Responses to Questions

**Comments to the Author**

1. Is the manuscript technically sound, and do the data support the conclusions?

Reviewer #1: Yes

Reviewer #2: Yes

2. Has the statistical analysis been performed appropriately and rigorously? 

Reviewer #1: Yes

Reviewer #2: Yes

3. Have the authors made all data underlying the findings in their manuscript fully available?

Reviewer #1: Yes

Reviewer #2: Yes

4. Is the manuscript presented in an intelligible fashion and written in standard English?

Reviewer #1: Yes

Reviewer #2: Yes

5. Review Comments to the Author

Reviewer #1: In this study, the authors investigated whether dogs can infer the causes of human success and failure in opening a door. In Experiment 1, dogs observed a competent versus an incompetent human, and in Experiment 2, they observed one door that could be opened and another that could not. The main finding was that dogs’ choices were random in both cases, suggesting that they did not infer either human competence or physical properties of the doors.

Overall, I found the study clearly reported and based on a simple and straightforward experimental design, which makes the results easy to interpret. The use of an accessible setup is a strength. At the same time, I felt that the novelty and impact of the work are somewhat limited, perhaps in part because the findings are negative results. Nonetheless, the study contributes to the growing literature on the limits of dogs’ causal reasoning and provides useful insights for future research designs.

I would like to raise a few potential concerns regarding experimental design and interpretation, which I hope may contribute constructively to the further refinement of this work.

【Major comments】

Main concern is the difference in what dogs were actually choosing in Experiments 1 and 2. In Experiment 1, the choice was between two “humans” who clearly differed in competence (one consistently succeeded vs. one consistently failed). In Experiment 2, however, the choice was framed as between two “gates,” but since dogs cannot open gates themselves, they are likely to have perceived this as a choice between the two humans standing next to the gates. Importantly, in this setup both humans had equivalent ability: each could open one of the gates but not the other. Thus, from the dog’s perspective, both humans were equally competent, and approaching either one would be equally reasonable. This raises the possibility that Experiment 2 did not truly test inference about the physical properties of the gates, but rather left the dogs with no basis to differentiate between the humans. I would encourage the authors to clarify this point in the Discussion, and perhaps consider alternative designs that could more cleanly separate choices based on door properties from those based on human agents. In addition, the way the Analyses are described (e.g., “comparing the proportion of trials for which dogs first approached the predicted gate” or “the amount of time dogs spent at each of the two gates”) reinforces the impression that dogs were choosing gates rather than humans even in Experiment1, which may be misleading given the actual task structure.

Another factor worth considering is the actual size of the doors used. Dogs are known to take their own body size into account when deciding whether to attempt passing through an opening (e.g., Lenkei et al., 2020, Animal cognition). Since many participants were large-sized dogs, if the doors were relatively narrow, this could have discouraged them from approaching or biased their choices, independently of the causal reasoning task. I could not find information about the door dimensions in the Methods; reporting this would clarify whether door size might have influenced the results.

In lines 336–362, the authors note that preliminary familiarization trials might alter the outcome. This point directly relates to findings by Kuroshima et al. (2017, Behavioural Processes), who showed that dogs can infer physical properties of objects (door weight) from human demonstrations only when they had prior personal experience with those objects. I suggest citing this study in that section, as it would strengthen the discussion by linking the present findings more explicitly to prior work. It would therefore be valuable to consider whether providing dogs with initial experience of openable vs. non-openable doors might have led to different results.

【Minor comments】

I think “Study�” should be unified as “Experiment2” (e.g., line 112,179,202).

Table1 and 2 might be okay to move to the supplement section.

The authors investigated the amount of time dogs spent at each of the two gates in each experiment. I noticed that the authors define “being close” as within 0.6 m of the agent or gate, but this information appears only in the Results section. It would be clearer to report this operational definition already in the Methods, together with a short rationale for choosing 0.6 m (e.g., based on prior studies or room setup). This would enhance clarity and reproducibility.

In the Discussion, the authors suggest that the results reported by Chijiiwa et al. (2022) might reflect simple mechanisms such as local or stimulus enhancement rather than genuine causal inference of human competence. I recommend citing the recent study by Jim et al. (2025, Animal Cognition), which raises very similar concerns in the context of dogs’ reputation formation. This would strengthen the connection to current debates in the field.

Reviewer #2: The authors investigated whether dogs can rationally infer the causes of failed actions.

They implemented an intelligent design in which dogs had to choose between two humans that were either not capable to open a gate due to incompetency or physical properties of the gate. I found this setup very elegant as it also ecological relevant in every-day lives of domestic dogs.

Dogs were unable to make rational inferences in both experiments. I find it essential to publish negative results to get the whole picture – in that case about the causal skills of dogs. As the also the methods seem to be appropriate, it is important to publish these data.

The thing I am not convinced regarding this paper is how the findings are embedded into the literature.

Whereas some papers seem to be a bit outdated (i.e. ref 15-17), also very relevant literature for the topic of the paper is missing:

On reputation

Nitzschner, M., Melis, A. P., Kaminski, J., & Tomasello, M. (2012). Dogs (Canis familiaris) Evaluate Humans on the Basis of Direct Experiences Only. PLoS ONE, 7(10), e46880. doi:10.1371/journal.pone.0046880

Kundey, S., De Los Reyes, A., Royer, E., Molina, S., Monnier, B., German, R., & Coshun, A. (2011). Reputation-Like Inference in Domestic Dogs (Canis familiaris). Animal Cognition, 14(2), 291-302. doi:10.1007/s10071-010-0362-5

On rationality

Kaminski, J., Nitzschner, M., Wobber, V., Tennie, C., Bräuer, J., Call, J., & Tomasello, M. (2011). Do Dogs Distinguish Rational from Irrational Acts? Animal Behaviour, 81(1), 195-203. doi:10.1016/j.anbehav.2010.10.001

On causal reasoning

Lampe, M., Bräuer, J., Kaminski, J., & Virányi, Z. (2017). The effects of domestication and ontogeny on cognition in dogs and wolves. Scientific Reports, 11690. doi:10.1038/s41598-017-12055-6

On reading human intentions

Schünemann, B., Keller, J., Rakoczy, H., Behne, T., & Bräuer, J. (2021). Dogs distinguish human intentional and unintentional action. Scientific Reports, 11(1), 14967. doi:10.1038/s41598-021-94374-3

The authors might also consider this paper, where dogs also have a choice between two humans after experience with them:

Silva, K., Bräuer, J., de Sousa, L., Lima, M., O’Hara, R., Belger, J., . . . Tennie, C. (2020). An attempt to test whether dogs (Canis familiaris) show increased preference towards humans who match their behaviour. Journal of Ethology, 38(2), 223-232. doi:10.1007/s10164-020-00644-4

Embedding these papers by rewriting the discussion will improve that paper. Overall, this is a good paper worth to be published in PLOS ONE but needs the above mentioned improvement in the theoretical parts, explaining the results.

Minor points: a videoclip of the procedure in the Supplementary Materials is always helpful.

6. PLOS authors have the option to publish the peer review history of their article (what does this mean? ). If published, this will include your full peer review and any attached files.

**Do you want your identity to be public for this peer review?** For information about this choice, including consent withdrawal, please see our Privacy Policy .

Reviewer #1: No

Reviewer #2: No

---

## [Author Response · Author response to Decision Letter 1]

9 Dec 2025

Reviewer #1:

In this study, the authors investigated whether dogs can infer the causes of human success and failure in opening a door. In Experiment 1, dogs observed a competent versus an incompetent human, and in Experiment 2, they observed one door that could be opened and another that could not. The main finding was that dogs’ choices were random in both cases, suggesting that they did not infer either human competence or physical properties of the doors. Overall, I found the study clearly reported and based on a simple and straightforward experimental design, which makes the results easy to interpret. The use of an accessible setup is a strength. At the same time, I felt that the novelty and impact of the work are somewhat limited, perhaps in part because the findings are negative results. Nonetheless, the study contributes to the growing literature on the limits of dogs’ causal reasoning and provides useful insights for future research designs.

> We thank the reviewer for their positive comments and for recognising the importance of contributing negative findings to the literature.

I would like to raise a few potential concerns regarding experimental design and interpretation, which I hope may contribute constructively to the further refinement of this work.

Major comments

Main concern is the difference in what dogs were actually choosing in Experiments 1 and 2. In Experiment 1, the choice was between two “humans” who clearly differed in competence (one consistently succeeded vs. one consistently failed). In Experiment 2, however, the choice was framed as between two “gates,” but since dogs cannot open gates themselves, they are likely to have perceived this as a choice between the two humans standing next to the gates. Importantly, in this setup both humans had equivalent ability: each could open one of the gates but not the other. Thus, from the dog’s perspective, both humans were equally competent, and approaching either one would be equally reasonable. This raises the possibility that Experiment 2 did not truly test inference about the physical properties of the gates, but rather left the dogs with no basis to differentiate between the humans. I would encourage the authors to clarify this point in the Discussion, and perhaps consider alternative designs that could more cleanly separate choices based on door properties from those based on human agents.

> We thank the reviewer for raising this thoughtful consideration. As the reviewer notes, across experiments, when making a decision, dogs were approaching both a particular gate and a particular experimenter. To be able to pass through the gate, (1) that gate must have been a functional gate that could be opened, and (2) the adjacent experimenter must have been a competent person capable of opening gates. Our particular manipulations in Experiments 1 and 2 were specifically designed to test whether dogs could reason about either or both causes for failure. For example, success in experiment 1 but not experiment 2 would suggest that dogs were sensitive to human competence but not the physical functionality of the gate, whereas success in experiment 2 but not experiment 1 would suggest the opposite sensitivity. Success in both experiments would suggest both sensitivities while failure in both experiments is consistent with a lack of both sensitivities.

>In each of the two experiments, we manipulated one of the factors (the competency of agents, or the functionality of the two gates) while holding the other constant. The value of this design is that it can tell us whether dogs are sensitive to one, both, or neither of these factors.

>We now clarify this in our introduction:

>“Crucially, although both human competence and gate functionality are needed for the gate to be opened, each of the two experiments manipulated only one of these factors. In doing so, we directly investigated whether dogs are sensitive to differences in human competency, object functionality, both factors, or neither factor.” (Lines 106-109)

>Nevertheless, we do agree with the reviewer that dogs’ dual failure in this study is difficult to interpret, and that if dogs are insensitive to object functionality, then they would not have perceived the manipulation in Experiment 2 as relevant. We have now outlined this possibility in our discussion section:

>“It is also possible that dogs are not attentive or sensitive to the functionality of gates, given that, although they can learn to do so [44], dogs don’t typically open gates themselves. If that is the case, then they may not have realised that both agents failed or succeeded to open one particular gate in Experiment 2. If that was the case, then either dogs may have perceived this task as choosing between two equally competent humans or they may not have noted any appreciable difference between either option. Either of these alternative hypotheses would predict that dogs should approach either of the gates at chance. Importantly, however, the former hypothesis would also have predicted that dogs would have discriminated between the competency of the two agents in Experiment 1, which they did not.

>Nevertheless, it might have been beneficial to provide additional scaffolding through which subjects could have understood how the gates worked. For example, we could have included additional demonstration at the start of each trial where an additional experimenter explicitly demonstrates the functionality of the gates to the dogs, perhaps by opening one gate to allow the dog access to their owner on the other side, but not the other. This familiarisation might allow any dogs that are unfamiliar with gates to learn that they could be opened for them. “Another option would be to allow subjects to pass through the gates once the experimenters opened them during the experimental demonstrations. Although we considered this possibility in the design of this study, we refuted it on three grounds: first, this would increase the length of time between demonstrations and moment when dogs made their choice at the end of the trial, potentially increasing existing cognitive load for the task; second, given the proximity of the gates to the owners, this may already provide dogs with proximity to their owners halfway through the trial and therefore reduce their motivation to approach them at the end; and third, and most importantly, differentially experiencing moving through either gate might lead dogs to make the correct choice at test not because of the competency of the agents, but because they learned to associate the “competent” agent with proximity to their owner in the social task, or the side of the “functional” gate with proximity to their owner in the physical task. In the physical task in particular, it is likely that this association would have led to a perseveration error, which dogs acquire after very few repetitions [45–47].

>“Alternatively, one could also envision an alternative to Experiment 2 where both gates are demonstrated by a single experimenter, who is repeatedly successful at operating only one gate but not the other. This would eliminate the dogs’ need to attend to the competencies of two different humans and from that infer the cause of their failure, as infants did in the Gweon and Schulz study [1], but it might have made the physical properties of the two gates more salient to subjects. Another option would be to train dogs to push gates open and then let them experience the functionality of both gates firsthand, given that previous research has shown that dogs can differentiate between “heavy” and “light” swinging doors, but only after they experience the two door types themselves [44].” (Lines 367-403)

>There is also another interpretation of the reviewer’s point that we agree deserves more discussion: since dogs are not acting on the gates themselves, they cannot solve the task simply by tracking whether each gate is functional. Instead, they must predict which human experimenter will be able to open which gate. When designing the task, we anticipated that this would not present an issue for dogs, given their familiarity with human social partners and pet gates, and since dogs almost invariably must rely on humans to open doors for them. In these senses, this experiment has a very high degree of ecological validity for pet dogs. Nonetheless, it remains the case that both of our experiments require dogs to recruit some amount of both social and physical cognition (e.g., even though Experiment 2 manipulated physical information, dogs still had to integrate that information with their expectations about how the experimenters would interact with those gates). It could be this combination of demands that hampered dogs’ performance, which we now address in our discussion:

>“Given that pet dogs are familiar with human social partners performing actions for them (including opening doors) and with pet gates, we anticipated that this task would be highly ecologically relevant for our subjects. However, since both our experiments require dogs to integrate both social and physical cognition to determine the cause of human or gate opening failures, it could be that these demands proved too challenging. Therefore, it is possible that although dogs may be capable of making inferences about causes for social or physical failures individually, the cognitive demands required to integrate information across both these domains in the context of our task may have been too high.” (Lines 404-411)

In addition, the way the Analyses are described (e.g., “comparing the proportion of trials for which dogs first approached the predicted gate” or “the amount of time dogs spent at each of the two gates”) reinforces the impression that dogs were choosing gates rather than humans even in Experiment1, which may be misleading given the actual task structure.

>We thank the reviewer for pointing out that the description of our analyses were confusing and we have now rephrased this section accordingly. Across experiments when making a choice, dogs were approaching both a gate and an experimenter, but we now clarify throughout the section that in Experiment 1 we were interested in measuring which of the two agents the dog approached, and in Experiment 2 we measured which of the two gates the dog approached:

>“Across experiments, when making a choice, dogs were approaching both a gate and a human experimenter but since we manipulated (and were interested in dogs’ sensitivity to) social competence in Experiment 1 and physical functionality in Experiment 2, we describe dogs’ choices as approaches to agents in Experiment 1 and to gates in Experiment 2.” (lines 231-235)

>“…our primary analysis was a Bayesian one-sample t-test comparing the proportion of trials for which dogs first approached the predicted agent or gate…” (Lines 237-238)

>“…where “Correct” is a 0 or 1 value indicating the dog’s choice of the causally predicted agent (Experiment 1) or gate (Experiment 2) in any given trial.” (Lines 242-243)

>“We additionally investigated the amount of time dogs spent at each of the two options (agents or gates) in each experiment using a Bayesian mixed-effects model. The dependent variable was the proportion of time spent at the predicted agent (Experiment 1) or gate (Experiment 2) over the total amount of time spent near both options, and the models are described by the equation…” (Lines 246-250)

Another factor worth considering is the actual size of the doors used. Dogs are known to take their own body size into account when deciding whether to attempt passing through an opening (e.g., Lenkei et al., 2020, Animal cognition). Since many participants were large-sized dogs, if the doors were relatively narrow, this could have discouraged them from approaching or biased their choices, independently of the causal reasoning task. I could not find information about the door dimensions in the Methods; reporting this would clarify whether door size might have influenced the results.

> We thank the reviewer for raising this important point, and for prompting us to provide additional details and discussion. The reviewer is correct that some dogs are hesitant to move through narrow spaces. We therefore checked that our participants were not fearful of gates and verified this before they participated in our experiment. In the training and procedures section of the manuscript, we note that “dogs were first allowed to investigate the study room in their own time, then encouraged to move through each gate twice in both directions” (Lines 156-157), and we now more fully describe this procedure in the methodology section too: “Dogs were only recruited for the study if they could be recalled by their owners and did not exhibit any fearful responses to indoor fencing or gating. This was confirmed when they arrived at the laboratory, when dogs were allowed to freely explore the testing room prior to the experiment, and in this case both gates were left open and dogs were encouraged to move through the two gates by their owners, twice clockwise and twice anticlockwise.” (Lines 122-127)

> We have also now included gate dimensions and referenced the Lenkei et al. (2020) study: “The fencing included two lockable pet gates (EveryYay steel gates) which could be swung open to form an opening measuring 46 cm x 69cm, which was sufficiently large for all participants to pass through, as dogs have been shown to avoid openings that are too small for their body size [36].” (Lines 129-132)

> Additionally, we have now added the breed size category for each of our participants into Tables 1 and 2.

In lines 336–362, the authors note that preliminary familiarization trials might alter the outcome. This point directly relates to findings by Kuroshima et al. (2017, Behavioural Processes), who showed that dogs can infer physical properties of objects (door weight) from human demonstrations only when they had prior personal experience with those objects. I suggest citing this study in that section, as it would strengthen the discussion by linking the present findings more explicitly to prior work. It would therefore be valuable to consider whether providing dogs with initial experience of openable vs. non-openable doors might have led to different results.

> We thank the reviewer for raising this important point and have now included the Kuroshima et al. (2017) study in that section of our discussion (Lines 367-403). The updated paragraphs are given as a response to the reviewer’s first comment.

Minor comments

I think “Study 2” should be unified as “Experiment 2” (e.g., line 112,179,202).

> We thank the reviewer for noting these inconsistencies and have now rephrased the three mentions of ‘Study 2’ as ‘Experiment 2’ on these three lines of the manuscript.

Table1 and 2 might be okay to move to the supplement section.

> We hope that the reviewer will be satisfied that we have opted to leave these tables in the main text, given that PLoS One does not impose a word limit to research articles, and that the sizes of dogs is quite relevant to the task, as the reviewer has pointed out in previous comments. We hope that by keeping both these tables within the main text, it will be even easier for the readers to access this information without having to find and download additional supplementary materials.

The authors investigated the amount of time dogs spent at each of the two gates in each experiment. I noticed that the authors define “being close” as within 0.6 m of the agent or gate, but this information appears only in the Results section. It would be clearer to report this operational definition already in the Methods, together with a short rationale for choosing 0.6 m (e.g., based on prior studies or room setup). This would enhance clarity and reproducibility.

> We have now clarified why this distance was chosen in the methods section: “This distance was chosen as it was slightly beyond the full extension of the gate when it was swung fully open, and so we considered any approach within the range of the open gate to be a choice for that side.” (Lines 135-138)

In the Discussion, the authors suggest that the results reported by Chijiiwa et al. (2022) might reflect simple mechanisms such as local or stimulus enhancement

---

## [Decision Letter · Decision Letter 1]

13 Jan 2026

Do dogs rationally infer the causes of failed actions?

PONE-D-25-40425R1

Dear Dr. Bastos,

We’re pleased to inform you that your manuscript has been judged scientifically suitable for publication and will be formally accepted for publication once it meets all outstanding technical requirements.

Kind regards,

Maria Eduarda Lima Vieira

Guest Editor

PLOS One

Additional Editor Comments (optional):

Dear authors,

I am pleased to announce that the revised version of the manuscript met the reviewers' expectations, and the manuscript is now accepted and ready for publication.

Reviewers' comments:

Reviewer's Responses to Questions

**Comments to the Author**

1. If the authors have adequately addressed your comments raised in a previous round of review and you feel that this manuscript is now acceptable for publication, you may indicate that here to bypass the “Comments to the Author” section, enter your conflict of interest statement in the “Confidential to Editor” section, and submit your "Accept" recommendation.

Reviewer #1: All comments have been addressed

2. Is the manuscript technically sound, and do the data support the conclusions?

Reviewer #1: Yes

3. Has the statistical analysis been performed appropriately and rigorously? 

Reviewer #1: Yes

4. Have the authors made all data underlying the findings in their manuscript fully available?

Reviewer #1: Yes

5. Is the manuscript presented in an intelligible fashion and written in standard English?

Reviewer #1: Yes

6. Review Comments to the Author

Reviewer #1: I would like to thank the authors for their thoughtful and detailed responses to my comments. The revised manuscript addresses my concerns very well, and the additional clarifications and discussion have greatly strengthened the paper. I have no further comments and am satisfied with the current version.

7. PLOS authors have the option to publish the peer review history of their article (what does this mean? ). If published, this will include your full peer review and any attached files.

**Do you want your identity to be public for this peer review?** For information about this choice, including consent withdrawal, please see our Privacy Policy .

Reviewer #1: No

---

## [Editor Report · Acceptance letter]

PONE-D-25-40425R1

PLOS One

Dear Dr. Bastos,

I'm pleased to inform you that your manuscript has been deemed suitable for publication in PLOS One. Congratulations! Your manuscript is now being handed over to our production team.

Kind regards,

on behalf of

Dr. Maria Eduarda Lima Vieira

Guest Editor

PLOS One